# Machine learning to extract muscle fascicle length changes from dynamic ultrasound images in real-time

Luis G. Rosa[1,2]*, Jonathan S. Zia[2,3], Omer T. Inan[2], Gregory S. Sawicki[1,4]

**1** School of Mechanical Engineering, Georgia Institute of Technology, Atlanta, Georgia, United States of America, **2** School of Electrical and Computer Engineering, Georgia Institute of Technology, Atlanta, Georgia, United States of America, **3** Emory University School of Medicine, Atlanta, Georgia, United States of America, **4** School of Biological Sciences, Georgia Institute of Technology, Atlanta, Georgia, United States of America

* lrosa3@gatech.edu

## Abstract

### Background and objective

Dynamic muscle fascicle length measurements through B-mode ultrasound have become popular for the non-invasive physiological insights they provide regarding musculoskeletal structure-function. However, current practices typically require time consuming post-processing to track muscle length changes from B-mode images. A real-time measurement tool would not only save processing time but would also help pave the way toward closed-loop applications based on feedback signals driven by *in vivo* muscle length change patterns. In this paper, we benchmark an approach that combines traditional machine learning (ML) models with B-mode ultrasound recordings to obtain muscle fascicle length changes in real-time. To gauge the utility of this framework for 'in-the-loop' applications, we evaluate accuracy of the extracted muscle length change signals against time-series' derived from a standard, post-hoc automated tracking algorithm.

### Methods

We collected B-mode ultrasound data from the soleus muscle of six participants performing five defined ankle motion tasks: (a) seated, constrained ankle plantarflexion, (b) seated, free ankle dorsi/plantarflexion, (c) weight-bearing, calf raises (d) walking, and then a (e) mix. We trained machine learning (ML) models by pairing muscle fascicle lengths obtained from standardized automated tracking software (UltraTrack) with the respective B-mode ultrasound image input to the tracker, frame-by-frame. Then we conducted hyperparameter optimizations for five different ML models using a grid search to find the best performing parameters for a combination of high correlation and low RMSE between ML and UltraTrack processed muscle fascicle length trajectories. Finally, using the global best model/hyperparameter settings, we comprehensively evaluated training-testing outcomes within subject (*i.e.*, train and test on same subject), cross subject (*i.e.*, train on one subject, test on another) and within/direct cross task (*i.e.*, train and test on same subject, but different task).

**Data Availability Statement:** All data for Rosa, Luis (2021), Machine Learning to Extract Muscle Fascicle Length Changes from Dynamic Ultrasound Images in Real-Time, are available in the Dryad

Digital Repository, via the following link: https://doi.org/10.5061/dryad.vq83bk3rp.

**Funding:** This work was supported in part by L.G. R.'s National Science Foundation NRT: Accessibility, Rehabilitation, and Movement Science (ARMS): An Interdisciplinary Traineeship Program in Human-Centered Robotics Award: 1545287 (https://www.nsf.gov/awardsearch/showAward?AWD_ID=1545287&HistoricalAwards=false), O.T.I.'s National Institute of Health, Institute of Biomedical Imaging and Bioengineering Award: 1R01EB023808 (https://grantome.com/grant/NIH/R01-EB023808-01), G.S.S.'s U.S. Army Natick Soldier Research, Development and Engineering Center Award: W911QY18C0140 (https://www.hsdl.org/?abstract&did=453253), and G.S.S.'s National Institute of Health's, Institute of Aging Award: R0106052017. The funders had no role in study design, data collection and analysis, decision to publish, or preparation of the manuscript.

**Competing interests:** The authors have declared that no competing interests exist.

## Results

Support vector machine (SVM) was the best performing model with an average $r = 0.70$ ±0.34 and average RMSE = 2.86 ±2.55 mm across all direct training conditions and average $r = 0.65$ ±0.35 and average RMSE = 3.28 ±2.64 mm when optimized for all cross-participant conditions. Comparisons between ML vs. UltraTrack (*i.e.*, ground truth) tracked muscle fascicle length versus time data indicated that ML tracked images reliably capture the salient qualitative features in ground truth length change data, even when correlation values are on the lower end. Furthermore, in the direct training, calf raises condition, which is most comparable to previous studies validating automated tracking performance during isolated contractions on a dynamometer, our ML approach yielded 0.90 average correlation, in line with other accepted tracking methods in the field.

## Conclusions

By combining B-mode ultrasound and classical ML models, we demonstrate it is possible to achieve real-time tracking of human soleus muscle fascicles across a number of functionally relevant contractile conditions. This novel sensing modality paves the way for muscle physiology in-the-loop applications that could be used to modify gait via biofeedback or unlock novel wearable device control techniques that could enable restored or augmented locomotion performance.

## 1. Introduction

In recent years, approaches employing machine learning (ML) techniques to help extract salient features from biomedical images have been rapidly growing to tackle the study and diagnosis of diseases spanning across human physiological systems [1, 2]. Researchers have successfully implemented ML based approaches to detect, classify, and measure volumes, lengths, and other spatial features of anatomical structures, such as the heart, brain, lungs, and muscles [3, 4]. Some advantages of these approaches over conventional manual measurements or semi-automated image extraction algorithms include increased throughput, improved clinical workflow, reduced health care costs, and the potential for real-time applications [5].

In clinical and basic physiology studies, B-mode ultrasound imaging has become the standard for measuring skeletal muscle architecture and dynamic length changes *in vivo* [6]. These 'under the skin' measurements of muscle anatomical structure and dynamic function have gained popularity because they can help elucidate mechanisms of muscle force production and energy use [7], that are difficult to uncover based on external measures due to decoupling between limb-joint and muscle dynamics [8–10]. As such, ultrasound has proven crucial for gaining insight into important physiological processes that are otherwise inaccessible through traditional physiological sensing modalities including electromyography, motion capture, and force ergometry. On the applied side, researchers have used ultrasound to study muscle structure-function in both healthy and pathological populations to understand physiological function and inform design of systems intended to provide movement assistance or rehabilitation [11]. Surprisingly, despite the widespread use of this imaging modality and the cumbersome and time-intensive nature of such manual approaches, hand-tracking images is still the gold standard for obtaining muscle fascicle lengths and length changes non-invasively [6]. Manual

tracking consists of going frame by frame and identifying the full extent of the selected fascicle [12]. This is a tedious task and hence many algorithms have been developed in an attempt to expedite and automate the process [13, 14]. Results from most of these tracking algorithms indicate B-mode ultrasound is a reliable and repeatable method for measuring muscle fascicle length changes in vivo [6, 13].

Despite current advancements, techniques are not yet widely available for ML based analysis of muscle physiology with ultrasound. There is work leveraging ML strategies for the analysis of specific skeletal muscle features [15–17], yet, to the best of our knowledge, most fail to measure dynamic fascicle length changes through time, or are limited for real-time implementation due to required processing times, often a consequence of more involved deep learning techniques [18]. Given ML's success in interpreting a wide range of biomedical images, we sought to apply ML to obtain real-time estimates of fascicle lengths from B-mode ultrasound images. Ultrasound images have a number of artifacts including attenuation, scattering, and refraction [15] that make automation, especially in real time, a formidable challenge. However, ML strategies are tunable and can be optimized for specific applications. Moreover, once trained, ML algorithm predictions can be ascertained rapidly, making *real-time* implementation feasible [19, 20].

One of the biggest challenges when implementing ML-based strategies is establishing a reliable and salient ground truth reference data set. Especially for physiological data—which tends to have a large amount of natural variation, the ground truth set should be robustly diverse. For human muscle data this manifests as a set of ultrasound image sequences recorded from multiple participants, performing an array of functional tasks that can be used to train and test ML-based image processing algorithms. To apply ML to extract muscle fascicle length estimates from ultrasound images, manual labeling could be used to establish the ground truth data set, but this would be tedious and extremely time consuming. Another approach is to leverage the state of the art automated tracking system to accelerate the process and obtain a 'ground truth' that is both accurate and can be rapidly (*e.g.*, in minutes not hours) established [21, 22]. UltraTrack is the most commonly used tool in the literature for muscle length analysis [6]. It leverages a Lucas-Kanade optical flow algorithm to track fascicles from one frame to the next [23], and it reduces processing time from hours of hand-tracking to anywhere from 5 to 40 minutes depending on the length of the study and on how accurately the experimenter implements its key-frame correction. This correction feature aims to minimize the drift effect seen as error accumulates from frame to frame. There are some accuracy limitations to UltraTrack as discussed in recent work [14] (especially in longer studies), yet due to its ubiquity and acceptable performance [6, 23], it seems an appropriate means of obtaining ground truth "on the spot" for a given user + participant and task. Having said this, the absence of real-time implementation capabilities remains a major limitation.

In this paper, towards developing a protocol that allows real-time measurement of muscle fascicle length changes, we benchmark a hybrid framework that combines ML and functional ultrasound to obtain real-time length changes from live images of human muscle in dynamic conditions. We show B-mode ultrasound images of soleus muscle during different locomotion-relevant tasks being fed to a number of different ML models yielding fascicle length estimates with low error and high correlation when compared to UltraTrack processed ground truth data. The resulting real-time, dynamic muscle imaging framework promises to not only speed up post hoc processing and analysis of ultrasound images of *in vivo* muscle contractile behavior, but also unlocks the possibility for novel muscle-in-the-loop applications that use muscle states to drive biofeedback training and/or assistive/rehabilitation devices [24].

## 2. Methodology

### 2.1. Experimental protocol

Six participants (3 male/ 3 female, age: 24 ± 4 years) voluntarily participated in this experiment after providing written informed consent (Georgia Institute of Technology IRB Protocol H17240). Participants had no current or previous significant lower-limb injury or gait pathology. B-mode ultrasound data were collected for five 24-second tasks (Fig 1) with the ultrasound probe (Telemed, ArtUs EXT-1H, LV8-5N60-A2 probe) wrapped to the right calf with 3M Vetrap Bandaging Tape to obtain images of soleus muscle fascicles (Fig 2A). The probe was aligned so that both aponeuroses were as close to horizontal as possible in the live ultrasound video feed. All data were collected by the same experimenter. Data were collected at 60 frames per second (fps); hence, each trial lasted approximately 24 seconds to reach the target 1400 frames we intended to capture. Note that on our computer (16 GB RAM), ultrasound software EchoWaveII (Telemed) limited maximum frame capture count per trial to 1500 frames. In each frame, we captured a single B-mode ultrasound image ("frame" and "image" are the same for our purposes and might be used interchangeably). (Fig 2A).

We selected five ankle movement tasks (Fig 1) to capture kinetic and kinematic conditions spanning high and low muscle forces and high and low muscle displacements; an array that captures muscle states experienced during natural movements (*e.g.*, locomotion: stance phase = high force, low displacement and swing phase = low force high displacement) [25]. For every participant the tasks were performed in the same order with at least 1 minute of rest in between each as follows: a) *constrained ankle flexion*: while seated, participants performed push-release loading cycles with their foot against the ground and no movement of their heels or knees, b) *free ankle dorsi/plantar/flexion*: while seated, participants plantarflexed and dorsiflexed over a full range of motion with their foot in the air, c) *calf raises*: while weight-bearing, participants cyclically raised and lowered their heels as far as they could d) *walking*: participants walked at 1.5 m/s on a treadmill (Tuff Tread, Inc.), and e) a *mix*: participants were encouraged to move their foot as desired while staying in place, but alternating weight-bearing and foot in the air. For tasks a-c, participants were encouraged to practice the timing so that they could perform around ~7 cycles of each task during the 24 seconds (3–3.5 Hz). Participants were offered breaks to rest between tasks, yet none required them.

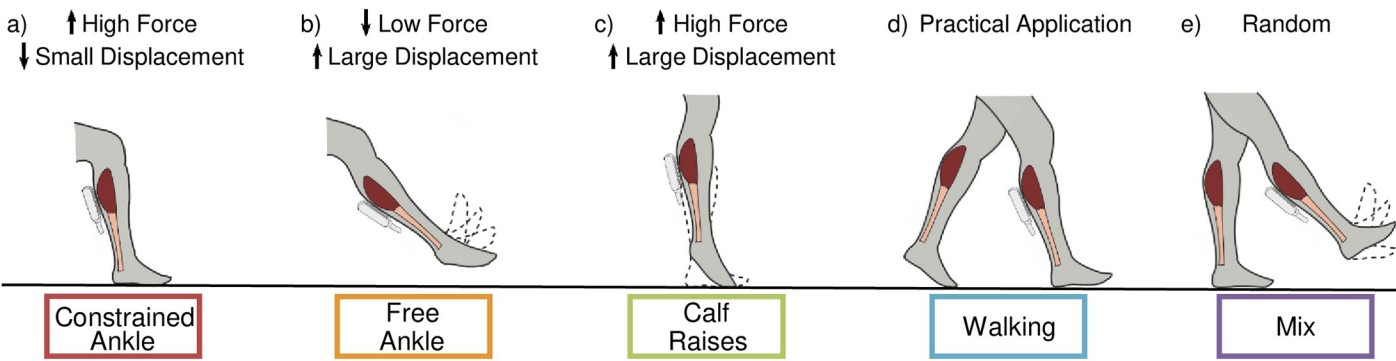

**Fig 1. Ankle movement tasks used to generate experimental data set of B-mode ultrasound images.** Dynamic B-mode ultrasound data were collected from the human soleus muscle for five ankle movement tasks with varying levels of force and displacement. Tasks were as follows: (a) (red) seated, constrained ankle plantarflexion with cyclic pushes of the foot on the ground and legs held still, (b) (orange) seated, free ankle plantar/dorsiflexion with foot in the air, c) (green) weight-bearing, calf raises with cyclic heel raises while standing in place, d) (blue) walking at 1.5 m/s on a treadmill, and e) (purple) a mix of directed movements while standing in place with foot on the ground and in the air. Tasks were specifically chosen to elicit combinations of high/low muscle fascicle force and displacement to provide a data set for training machine learning (ML) algorithms that had a high degree of variability. Task color coding is consistent throughout the paper.

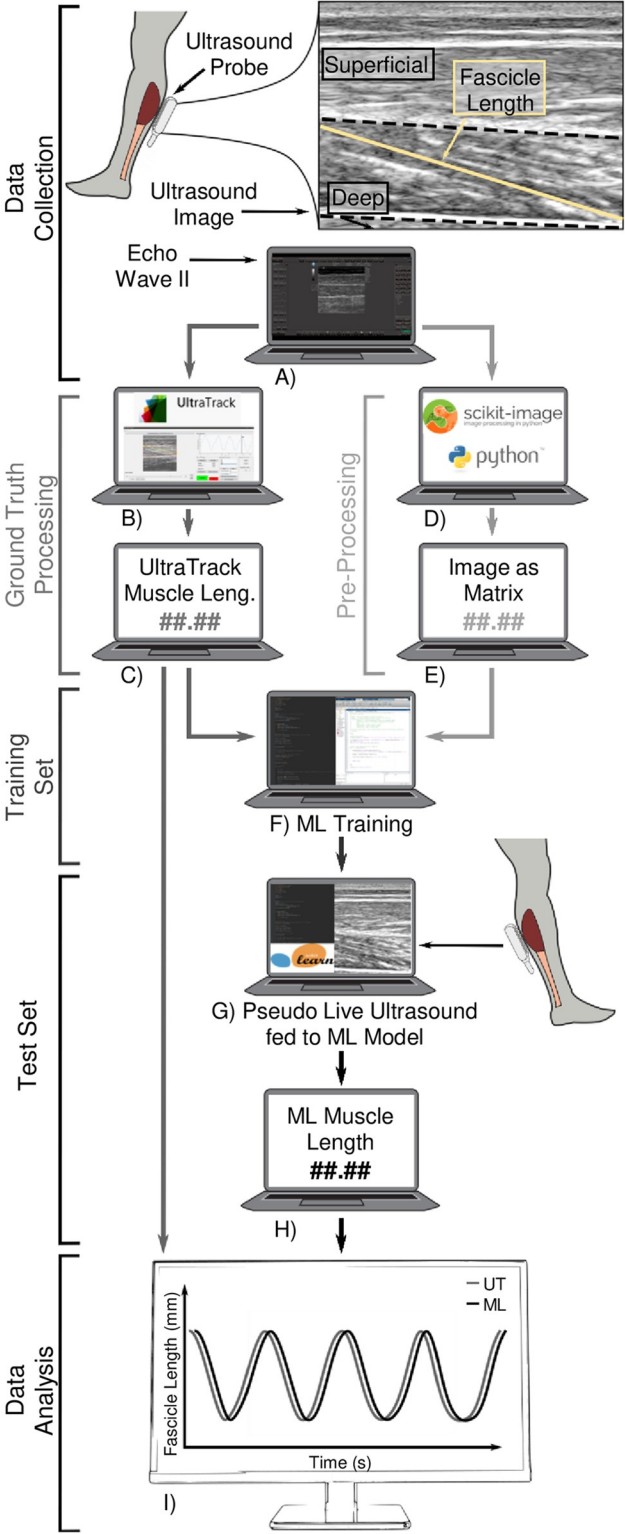

**Fig 2. Workflow from B-mode ultrasound image acquisition and processing to machine-learning (ML) model training and real-time muscle length tracking.** A) In each frame, a B-mode image containing human soleus muscle fascicle (yellow line) is captured via ultrasound probe. B-C) Training and ground truth muscle fascicle length change data sets are obtained via UltraTrack software. D-E) Images in each frame are cropped and down-sampled using Python code to implement open-source functions. F) A machine learning (ML) model is trained using outputs from C

and E. G) Unique images not seen by the ML model in the training stage are fed into the ML model. H) The ML model yields pseudo real-time muscle fascicle length measurements. I) Performance metrics are calculated (e.g., RMS error and correlation), to compare ML-derived measurements to the ground truth.

## 2.2. Data processing

**2.2.1. Muscle fascicle length ground truth.**   We used UltraTrack tracking software [21] to process B-Mode images and define a fascicle length in each of the 1400 frames from every participant performing each task (Fig 2B). We used UltraTrack's automated fascicle tracking and added key frame correction whenever appropriate to reduce drift at least twice per set to ensure there were no outstanding errors. We used the first 1000 out of those 1400 images as our muscle fascicle length *training set* (Fig 2C–2F) and the remaining 400 images to define a ground-truth *testing set* for verifying performance (Fig 2C–2I). Traditionally, machine-learning (ML) model accuracy is assessed using random samples rather than the last portion of a data set. The approach we chose here, to select the last portion of data set as ground-truth, was done intentionally, to emulate the process that would be undertaken if we applied the framework to obtain real-time measurements in a mobile laboratory setting.

**2.2.2. Ultrasound image pre-processing.**   To reduce computational requirements and increase processing speed, we cropped and down-sampled B-mode images into smaller matrices (**I in** S1 File). First, we cropped them to focus on the soleus muscle, the largest of the calf muscles and often a target for clinical intervention [26], and eliminated both superficial (*i.e.*, gastrocnemius muscle) and deep regions (Fig 2D) of the image. We then down-sampled the images at four different reduction rates using the "block_reduce" function from Scikit-Image, an open-source image processing library for Python [27]. Down-sampling the 658 x 556 pixel images yielded pixel matrices of 4 different sizes corresponding to each of the four down-sampling rates, from we chose the heaviest down-sampling which yielded the smallest matrices per image (**S1 Table in** S1 File).

**2.2.3. Machine-learning (ML) model implementation.**   To train our machine learning (ML) models (Fig 2F), we used 1) the first 1000 pixel brightness matrices representing the first 1000 images as our training data input (Fig 2E), and 2) the corresponding UltraTrack derived muscle fascicle lengths for those 1000 images as our ground truth (Fig 2C). Once trained, we inputted the remaining 400 matrices/frames from the test set to the ML model to estimate the respective muscle fascicle lengths based on what the model learned from the first 1000. These 400 frames were fed sequentially (in the same order as captured) to emulate how the computer would process real-time ultrasound images (Fig 2G) which yielded an output of pseudo-real-time soleus muscle fascicle length estimates (Fig 2H). To clarify, we define pseudo-real time to acknowledge that the data we ran through the workflow rubric described in Fig 2 was done post-hoc in this instance. This was to allow us to examine and optimize multiple combinations of algorithms and parameter settings since the actual real-time implementation cannot be validated in real-time non-invasively. Note that the suggested technique although capable of producing real-time measurements as shown in S3 Video, will be limited mainly by hardware and ML processing of B-Mode images (**III in** S1 File).

**2.2.4. Machine-learning (ML) model and parameter optimization.**   Parameter settings within a given machine-learning (ML) model can be tuned to optimize input-output performance. We examined five ML models, both linear and non-linear, that were available in the Scikit Learn library [28]: Lasso, Ridge, Linear Support Vector Regression (Linear SVR), Epsilon-Support Vector Machine (SVM) with 'rbf' kernel, and Random Forest. For each ML model we used the grid search method to test up to ~100 unique parameter combinations and find optimal performance (**II in** S1 File). To optimize, we found parameter settings that

simultaneously maximized correlation and minimized root mean square error (RMSE) of ML processed vs. UltraTrack processed (*i.e.*, ground-truth) muscle fascicle length change data. In other words, we formulated an objective function where correlation and RMSE had equal, non-dimensional weighting. Since correlation values span from 0 to 1 and RMSE values span from 0 to variable numbers, we scaled the obtained RMSE values for each optimization to span a 0 to 1 range. That is, we normalized the RMSE values by dividing absolute RMSE by the largest observed across participants and tasks (= ~14 mm). Then, with both correlation and RMSE on the same scale, we simply added the difference of 1 –the adjusted RMSE (to make lower values better) to the correlation value and selected the hyper parameters within each ML model that yielded the largest sum (See Sec. 3.4 for application dependent changes to this formulation). Once optimal hyperparameters for each ML model were selected, we were able to compare ML model performance across conditions. Note that we also used this objective function to select the best down-sampling rate and found that the heaviest down-sampling yielded the best correlation/RMSE balance (**I in** S1 File). Hence, the results presented across ML models are all from images reduced to matrices at the same lightly optimized rate (Fig 2E).

## 2.3. Machine-learning (ML) model evaluation

We selected Pearson's Correlation Coefficient (*r*) and root-mean square error (RMSE) as our two main evaluation metrics as they are regularly employed to assess accuracy in both ML and physiology literature [13, 29]. Because our ML models always yielded the exact same results when under the same training/testing conditions (*i.e.*, deterministic input-output behavior), the common Coefficient of Multiple Correlations (CMC) and Intraclass Correlation Coefficient (ICC) were not appropriate in this study. We instead focused on how the *r* and RMSE values changed under different train-test scenarios and how these compared to accepted ranges for 'good' performance from the literature. Generally, although application dependent, we consider $r \leq 0.35$ as weak, 0.36 to 0.67 as moderate, and 0.68 to 1 as strong or very strong correlation [30, 31]. The utility of RMSE largely depends on whether/how it is normalized to account for the range over which the measurement can naturally vary. As a benchmark for performance, we quantified how our RMSE in absolute muscle fascicle length compared to the existing literature examining tracking of images containing fascicles of similar lengths (*e.g.*, human soleus = ~30–50 mm at rest), which tend to have RMSEs under 10% of the mean length [13, 14]. In addition, for qualitative comparison, we examined how these ML estimates compared to the 'ground-truth' (= UltraTrack processed) in 'real-time' play-back mode by visualizing both signals simultaneously on a monitor to mimic what a user might see during biofeedback applications (Fig 2I).

Using RMSE and Pearson's Correlation Coefficient (*r*), we evaluated three potential muscle fascicle B-mode ultrasound imaging data *train-test* use cases: a) *direct training*: in which we trained the ML model and then tested it using data from the same participant and the same task (*e.g.*, train on participant #3 walking data to measure participant #3 during walking), b) *cross-participant training*: in which we trained the ML model with data from a different participant than the one being tested (*e.g.*, train on participant #6 walking data to measure participant #3 during walking), and c) *cross-task training*: in which we trained the ML model on data from the same participant but from a different task than the one being tested (*e.g.*, train on participant #3 using heel raises to measure participant #3 during walking). Note that we used the cross-task condition to measure only walking. We used the same train-test analysis protocol for direct, cross-participant and cross-task cases in which, of 1400 total image frames per condition, we used the first 1000 frames to train to 'ground truth', then the last 400 frames to test

versus 'ground truth'. Testing against ground truth meaning we calculated the correlation and RMSE values between the 400 ground truth frames and the ML-model-predicted 400 frames.

## 3. Results and discussion

In this paper, for the first time, we were able to demonstrate the feasibility of ML algorithms for obtaining accurate muscle fascicle length measurements from B-Mode ultrasound images in real-time (**III in** S1 File, S3 Video). After optimizing each ML model (Lasso, Ridge, Linear SVR, SVM, Random Forest) to simultaneously maximize Pearson's correlation, *r* and minimize RMSE with equal weighting across all movement tasks, we obtained the best performance using SVM for both direct and cross participant training scenarios (Table 1). Note that we did not implement a neural network since after preliminary testing non-linear models like Random Forest, we did not see a significant performance increase. SVM yielded an average correlation (*r*) of 0.70 ±0.34 with an average RMSE of 2.86 ±2.55mm when optimized for all direct training conditions (Table 1a) and yielded an average correlation of 0.65 ±0.35 with an average RMSE of 3.28 ±2.64mm when optimized for all cross-participant movement tasks (Table 1b). Note that correlation value averages in all Tables are averaging all the corresponding sub-components (e.g., in Table 1a we obtained correlation values for each of the models by averaging the correlations obtained for every subject and every task with direct training for that model). These results show moderate to strong correlation values comparable to previous reports documenting muscle fascicle length tracking algorithms. Indeed, our result demonstrates RMSE of ~3mm or ~6% of the median fascicle length (= ~50mm) for most trials, which is in line with the 10% RMSE seen in related studies [6, 13, 14]. It is important to note that we obtained similarly strong results from all five ML models, indicating that, if properly tuned, many ML models have potential to generate accurate tracking of muscle fascicle length from B-mode images without requiring deep learning. Given its overall best performance, we focus our analysis on direct, cross-participant, and cross-task training for the optimized SVM ML model *only*.

### 3.1. Direct training

From direct training we obtained correlation averages of 0.90 for both the free ankle and calf raises, 0.73 for constrained ankle, 0.52 for walking, and 0.41 for the random mix, all while having RMSEs less than 4mm (Table 2a). Note that these values represent averages and that there is considerable spread. We would recommend visual inspection of the results to help confirm their usability. Pseudo real-time estimates as they would have been obtained if the system were applied in real-time for one subject indicate that even in cases of moderate or low correlation (<0.6) and/or higher RMSE (>4mm), it is evident that the ML algorithm *is able* to

**Table 1. Machine learning (ML) model comparisons.**

| | a. Direct Training Averages per Model | | b. Cross-Participant Training Averages per Model | |
|---|---|---|---|---|
| | *CORR (r)* | *RMSE (mm)* | *CORR (r)* | *RMSE (mm)* |
| *Lasso* | 0.68 ±0.37 | 2.76 ±2.38 | 0.50 ±0.35 | 2.72 ±2.82 |
| *Ridge* | 0.68 ±0.29 | 2.77 ±2.35 | 0.65 ±0.29 | 3.31 ±2.78 |
| *LinearSVR* | 0.69 ±0.32 | 2.73 ±2.30 | 0.55 ±0.30 | 2.73 ±2.72 |
| *SVM* | 0.70 ±0.34 | 2.86 ±2.55 | 0.65 ±0.35 | 3.28 ±2.64 |
| *Random Forest* | 0.60 ±0.29 | 3.65 ±2.91 | 0.49 ±0.30 | 3.44 ±2.21 |

Correlation is unitless, shown as CORR *(r)*, Root Mean-Square Error (RMSE) shown in millimeters. Average correlations and RMSEs were obtained averaging all corresponding subjects and tasks.

**Table 2. Ankle movement task comparisons for SVM across training schemes.**

| | *a. Direct Training Averages per Task* | | *b. Cross-Participant Training Averages per Task* | | *c. Cross-Task Training Averages for Walking* | |
|---|---|---|---|---|---|---|
| | *CORR (r)* | *RMSE (mm)* | *CORR (r)* | *RMSE (mm)* | *CORR (r)* | *RMSE (mm)* |
| *Constrained Ankle* | 0.73 ±0.34 | 0.84 ±0.58 | 0.62 ±0.33 | 0.85 ±0.66 | 0.20 ±0.19 | 2.45 ±0.89 |
| *Free Ankle* | 0.90 ±0.32 | 3.67 ±3.01 | 0.81 ±0.35 | 4.31 ±3.14 | 0.19 ±0.17 | 2.19 ±0.92 |
| *Calf Raises* | 0.90 ±0.30 | 2.39 ±1.19 | 0.89 ±0.33 | 2.56 ±1.31 | 0.24 ±0.16 | 2.47 ±1.13 |
| *Walking* | 0.52 ±0.22 | 2.08 ±1.45 | 0.29 ±0.18 | 2.22 ±1.38 | - | - |
| *Mix* | 0.41 ±0.26 | 2.45 ±3.37 | 0.38 ±0.26 | 3.19 ±3.48 | 0.26 ±0.21 | 2.32 ±1.20 |
| *Average* | **0.69 ±0.29** | **2.29 ±1.92** | **0.60 ±0.29** | **2.63 ±1.99** | **0.22 ±0.18** | **2.36 ±1.03** |

Correlation is unitless, shown as CORR *(r)*, Root Mean-Square Error (RMSE) shown in millimeters. Average correlations and RMSEs were obtained averaging all corresponding subject combinations.

qualitatively track muscle fascicle lengths well, capturing major features like peaks and inflection points in time-series data (Fig 3).

We analyzed direct training scenarios separately with an eye toward using UltraTrack as a rapid means of obtaining ground truth for the ML model to be trained for specific movement

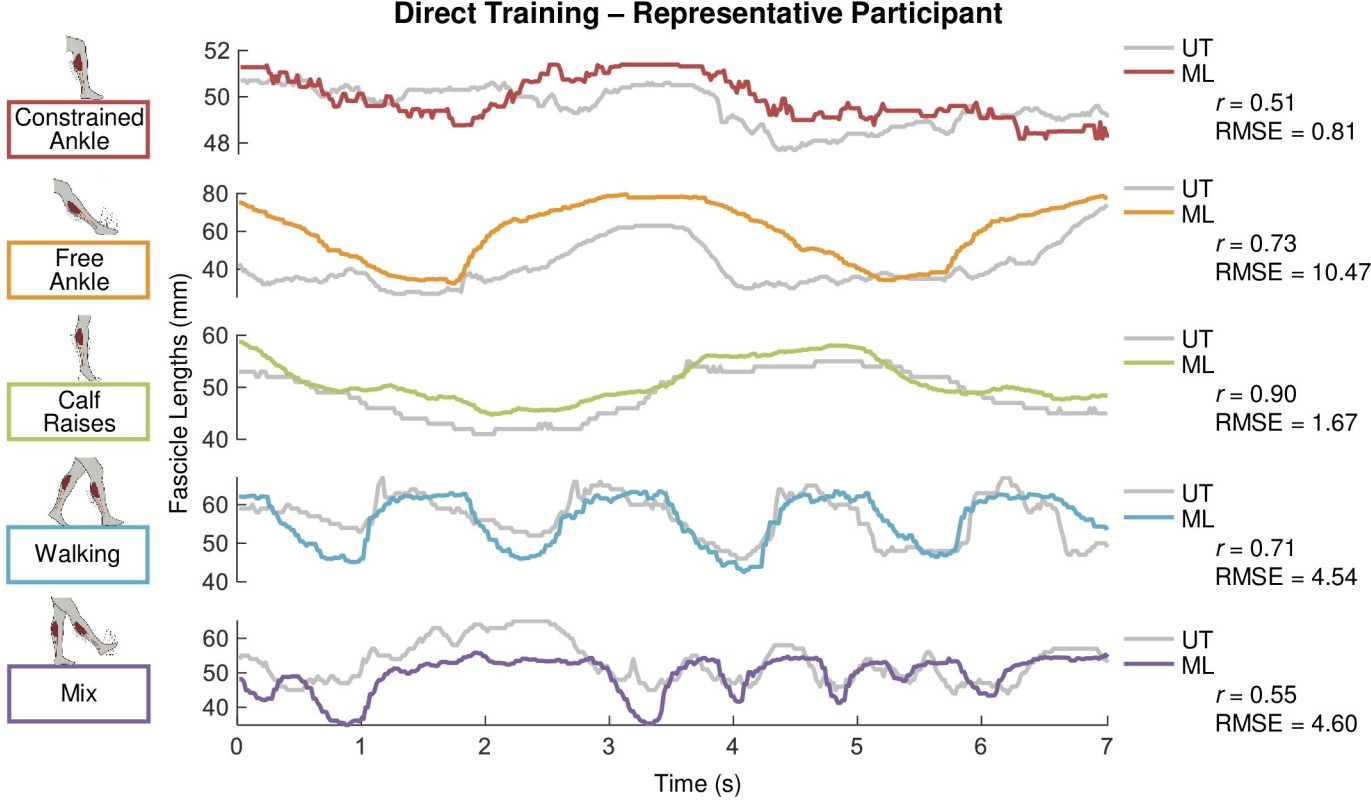

**Fig 3. Muscle fascicle length versus time from optimized ML algorithm with direct training across tasks.** Subplots show raw output of muscle fascicle length (mm) versus time (seconds) estimated using B-mode ultrasound images fed through a support vector machine (SVM) machine-learning (ML) model with optimized hyperparameters for a representative participant X during several cycles of a given ankle movement task (e.g., constrained ankle = red (top); mix = purple (bottom)). In each case, the SVM ML output was generated following a training procedure using image data from the same participant X and same task as that used to measure muscle fascicle length in the test participant X (i.e., direct task to task training). Muscle fascicle lengths derived from inputting the same B-mode images into UltraTrack (UT) software (gray) are included as 'ground-truth' to give context regarding correlation and RMSE across tasks. Note that scaling of Y-axes differs from panel to panel.

tasks related to locomotion. The process of setting up the ultrasound probe, collecting B-mode ultrasound data in each movement task, processing data in UltraTrack and then training the ML model onsite so that it is ready to make real-time measurements was on average accomplished in approximately 20–30 minutes. This opens opportunities for researchers to train their ML models on a specific individual patient / participant in a specific movement task right before the experiment or rehabilitation session starts. This innovation could potentially eliminate the need for robust one-size fits all solutions that work between different sessions, individuals, and movement conditions.

### 3.2. Cross-training

From cross-*participant* training we obtained correlation averages >0.80 for both free ankle and calf raises (0.81 and 0.89 respectively), 0.62 for constrained ankle, 0.29 for walking, and 0.38 for the random mix movement tasks while having RMSEs less than 4mm for *all* except the free ankle condition (Table 2b). We believe cross-participant training is successful in part due to the magnitude of the pre-processing down-sampling which likely removes unnecessary details and differences between individuals while keeping salient information needed to extract accurate muscle fascicle length measurements. As with direct training, even in scenarios like walking, where formal correlation values are not very high (<0.7), ML derived muscle fascicle length measurements qualitatively follow the ground truth data across all participants. (Fig 4).

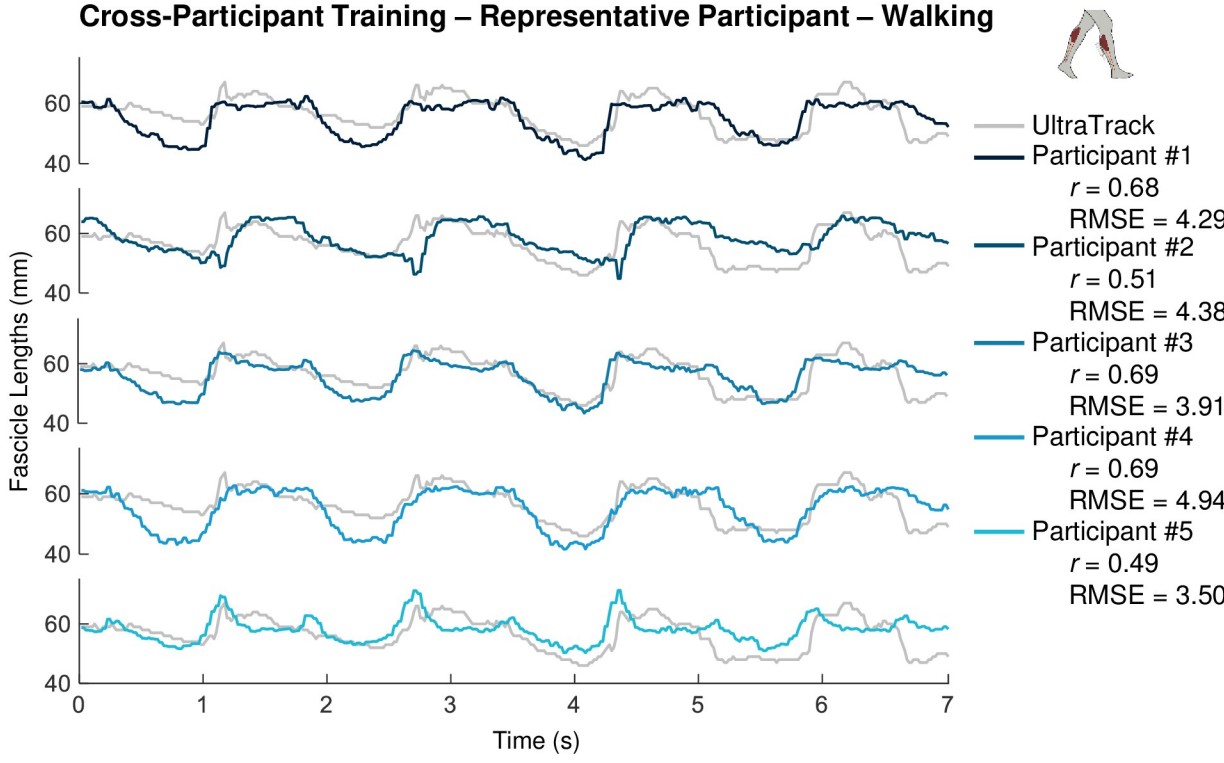

**Fig 4. Muscle fascicle length versus time from optimized ML algorithm with cross-participant training in the walking task.** Subplots show raw output of muscle fascicle length (mm) versus time (seconds) estimated using B-mode ultrasound images fed through a support vector machine (SVM) machine-learning (ML) model with optimized hyperparameters for a representative subject X during several cycles of the walking task. In each case, the SVM ML output was generated following a training procedure using image data from walking in each of the other participants (e.g., participant #1 = dark blue, participant #5 = light blue) and then used to measure muscle fascicle length in the test participant X (i.e., cross participant training). Muscle fascicle lengths derived from inputting the same B-mode images into UltraTrack (UT) software (gray) are included as 'ground-truth' to give context regarding correlation and RMSE across tasks.

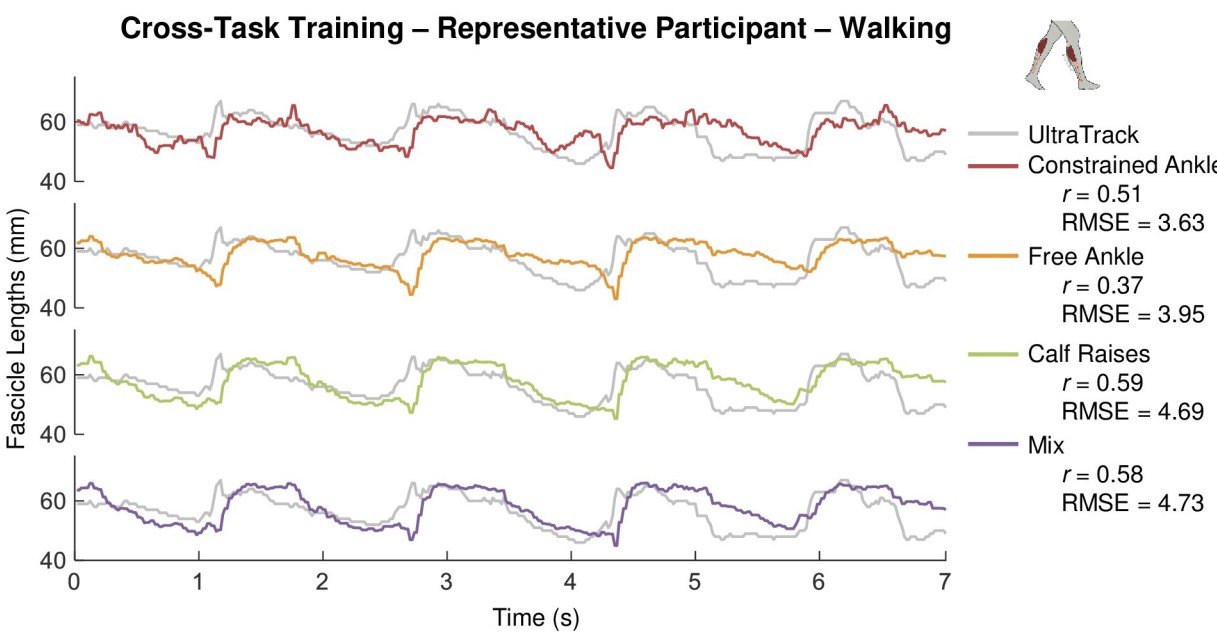

**Fig 5. Muscle fascicle length versus time from optimized ML algorithm with cross-task training in the walking task.** Subplots show raw output of muscle fascicle length (mm) versus time (seconds) estimated using B-mode ultrasound images fed through a support vector machine (SVM) machine-learning (ML) model with optimized hyperparameters for a representative participant X during several cycles of a given ankle movement task (e.g., constrained ankle = red (top); mix = purple (bottom)). In each case, the SVM ML output was generated following a training procedure using image data from each of the other tasks (e.g., constrained ankle = red, mix = purple) and then used to measure muscle fascicle length during walking in the test participant X (i.e., cross-task training). Muscle fascicle lengths derived from inputting the same B-mode images into UltraTrack (UT) software (gray) are included as 'ground-truth' to give context regarding correlation and RMSE across tasks.

From cross *task* training we obtained an average correlation of 0.22 ±0.18 and average RMSE of 2.36 ±1.03mm when measuring walking data after training our SVM ML model with data from the other movement tasks (Table 2c). These correlation values are lower than direct or cross-participant training since each task covers a different combination of ankle joint range of motion, soleus muscle loading, and overall noise. With under 20 seconds of training data (1000 frames), it was challenging to generalize training and make accurate muscle fascicle length measurements in walking. However, we observe once again that despite low formal correlations (<0.6) our ML based tracking still generates useable data that captures qualitative features of ground truth quite well. (Fig 5). After a closer look at our data, we observe that when compared to cross-participant or direct training, correlation averages for cross-task training are much lower due to a more frequent lack of correlation. Thus, despite potential convenience, there is a larger risk of poorer muscle fascicle length measurements when employing cross-task training vs. the other approaches (Table 2).

In practical terms, our data suggest that researchers using conventional ML models (*e.g.*, SVM, Lasso etc.) to extract muscle fascicle length measurements should aim to train their model on each individual and on the movement task that they intend to measure (*i.e.*, direct calibration) whenever possible if high accuracy is required. We note, however, that the despite trading-off time savings or out-of-lab use for loss of accuracy (*i.e.*, lower correlation or RMSE), cross-training might still be a viable option depending on the demands of the specific application (**see** Table 2 **and Sec. 3.4**). Interestingly, our data indicate that ML algorithms appear to be more sensitive to changes in muscle fascicle displacement magnitude and/or loading (*i.e.*, across ankle movement tasks) (Fig 3) than changes in the exact muscle features distinctive of each individual (*i.e.*, across participants) (Fig 4). This from the significantly higher correlations

cross-participant vs cross-task (Table 2). Hence, when given the choice, cross-participant training is preferable over cross-task training.

### 3.3. Muscle fascicle length tracking performance

When applying ML to extract information from data, it is important to consider differences between the 'ground truth' used for training the algorithm and testing its measurement accuracy versus the underlying objective state of the desired sample, in this case muscle fascicle lengths. The 'ground-truth' which most muscle fascicle tracking algorithms are tested against uses data that are hand-tracked in each image frame. Although hand-tracked imaging data still does not perfectly capture actual *in vivo* fascicle lengths, it is still considered the most accurate, non-invasive measurement method [13]. In this study, we defined ground truth using Ultra-Track, an automated tracking software package, which likely introduces additional error when compared to hand-tracked fascicle lengths. UltraTrack uses a Lucas-Kanade optical flow algorithm that inherently accumulates error from frame to frame [21, 23]; hence, longer data trials and/or non-cyclical movement where the key frame correction tool is not as effective, promote larger error accumulation [14]. This error accumulation is a larger factor towards the end of a given trial, which is particularly relevant in our study, given that we used the last portion of our data streams to test our ML models. (*i.e.*, last 400 of 1400 frames). These errors likely impacted the training of our ML models and thus, are no doubt reflected in our reported correlations and RMSEs, which would be different for a hand tracked ground truth. Yet, as mentioned in Section 2.2.1, evaluating the last portion as a separate continuous set better emulates how frames would be processed during real-time applications. Furthermore, it enables qualitative evaluation of the resulting curves to compliment correlation and RMSE.

On the other hand, despite inducing potential errors, UltraTrack is the most widely used semi-automatic muscle fascicle length tracker [6] and it provides the means to rapidly obtain a ground truth. Indeed, using UltraTrack as ground truth in this study allowed us to demonstrate the feasibility of collecting a test B-mode ultrasound imaging data set, rapidly tracking that data onsite, using it to train the ML model, and then applying the optimized ML model to process novel images in real-time, all during a single data collection session in the lab (Fig 2, **S3 Table in** S1 File). Beyond real time applications, this trade-off of speed for accuracy could enable offline batch processing of data from more participants and movement tasks on much shorter timescales when compared to hand-tracking.

Our data demonstrate that the correlation between UltraTracked and ML derived muscle fascicle lengths depends on the task (Fig 3, Table 2a). Movement tasks that did not involve explicit ground contact impacts (constrained ankle, free ankle, calf raises) yielded $r = 0.84$ on average for direct training, while more dynamic tasks with more variable tissue loading (walking, mix) had $r = 0.47$ (Table 2a). A deeper look into our data, suggests that the most dynamic tasks may worsen the accumulation of error in UltraTrack as described in the previous paragraph; hence, correlation values in these cases may be lower due to a drifted ground truth [13] in the last 400 frames used for testing accuracy compared the first 1000 frames used for training. Most of the literature using the same algorithm as UltraTrack presents measurements of muscle fascicle length changes from highly controlled contractions on dynamometers [6, 14, 23], with more repeatable muscle forces and ankle motions. These idealized data likely exhibited less 'noise' in the B-mode ultrasound images and better leveraged the "Key Frame Correction" feature in UltraTrack thereby facilitating more accurate fascicle tracking. Furthermore, studies that analyze muscle fascicle length tracking accuracy in dynamic tasks such as walking typically break down the data by stride, creating manageable chunks with little error accumulation [6, 22]. In short, it is important to compare results from our ML model-based muscle

fascicle tracking with data from equivalent movement tasks. For example, our calf raises can be compared to dynamometer studies that also applied high forces in relatively static ankle postures. In calf raises, with direct training, our ML model yielded 0.90 average correlation, directly in line with the similarly high correlations reported for muscle fascicle length measures taken from hand tracked or UltraTracked images recorded in dynamometer studies in the field [13, 23, 30].

## 3.4. Limitations, areas for improvement and potential applications

The ability to non-invasively measure human muscle fascicle lengths in real-time using ultrasound could pave the way toward novel 'muscle-in-the loop' biofeedback paradigms and wearable device control schemes for rehabilitation or augmentation of human movement. In the current work, we demonstrate that employing artificial intelligence to process ultrasound images is a key step in this direction. Indeed, ML algorithms, like SVM are adaptable and can be rapidly trained and optimized to extract ground truth data with high correlation and low RMSE in real-time.

Despite successfully benchmarking correlation and RMSE values from ML-derived muscle fascicle length tracking that compare well with other standardized tracking approaches (**S3 Table in** S1 File), we believe demonstrating the utility of our application in a real-world application will be the ultimate test of what is "good enough". Different applications will fall into different categories along the continuum of processing speed-accuracy trade-off, and this will ultimately drive new innovations. So far, ultrasound–based muscle fascicle length measurements have been applied predominantly in *post hoc* analysis of muscle function during movement, and this has driven a hyper-focus on improving measurement accuracy and reliability without much pressure on improving the speed with which measurement can be taken.

Our results here should help pave the way for real-time 'muscle in-the loop'-approaches that will help redefine application-specific optimization of image tracking reliability and accuracy needs. For example, it may be that in some applications, accuracy is paramount only during a certain phase of the movement, bringing into focus a new set of factors in the image processing pipeline that will need to be optimized. In these cases, ML model parameters might be tuned to provide either higher correlation or lower RMSE rather than equally balancing these priorities. In addition, closed-loop applications may not need accurate (in mm) muscle length measurements at all. Instead, biofeedback applications designed to steer muscle length change *in vivo* (*e.g.*, to avoid rapid stretching) might probe other kinds of muscle state information such as tracking the relative (*i.e.*, unitless) peak or average length /velocity with respect to a within-participant threshold designed to optimally evolve over time.

It is instructive to consider specific use cases in order to establish the accuracy needed from the ML model used in the image processing pipeline. One example use-case we have considered is to use the ML-driven approach outlined in this paper to implement a muscle in-the-loop biofeedback system aimed at minimizing the metabolic energy cost of individual leg muscle contractions. A reduction in metabolic cost of walking or running on the order of ~10% could have significant impact on mobility and quality of life [32]. Furthermore, recent studies have shown that average operating length of a muscle actively generating force can impact its energy consumption. Roughly speaking, for the human soleus, a 1% change in muscle length corresponds to a ~1% change in metabolic cost [24, 26]. This relationship offers a window into the resolution that would be needed to enable a user wearing a real-time ultrasound imaging device to gain closed-loop control of their own muscle length. Similarly, muscle strain rates above 30% are known to cause injuries [33, 34], and this physiological fact could be used to guide the resolution needed in an image-based feedback system that maintains muscle length

below safety thresholds. It is interesting to note, that because muscle performance and injury propensity is often couched using relative rather than absolute measures of fascicle length (i.e., strain vs. mm), using ML models to track lengths of longer muscles would reduce the bearing of inaccuracies in data extraction. Furthermore, feeding ultrasound images along with data from other sensing modalities (*e.g.*, electromyography [35], tensiometry [36, 37]) thru ML schemes could provide improved measurement robustness and /or resolution.

It is important to note that our approach to optimizing the ML-based estimates of muscle fascicle length changes from B-mode images was not exhaustive, but instead represents a first step toward the long-term goal of developing optimized muscle-level feedback systems for augmented movement. For example, the amount of data we used for training, 1 subject and 1 condition (1000 frames), is minimal in comparison to what most traditional convolutional neural networks use, yet it shows how well a simple lab-applicable approach performs. This is especially useful for researchers with limited access to data and/or limited deep learning experience. Nonetheless, some ways of improving our ML model based predictions include: 1) improving ground truth quality by considering more accurate alternatives to UltraTrack, 2) optimizing the size and image quality of the training data set 3) implementing real time filtering and signal processing techniques (*e.g.*, Kalman, etc.), 4) using more capable imaging hardware (*e.g.*, increasing ultrasound sampling frequency in time, reducing latency, etc.), 5) optimizing feature level aspects of the image processing through time (*e.g.*, increasing resolution at a specific gait phase), 6) devising movement tasks that can generate training sets that robustly generalize for the intended application (*e.g.*, strenuous and/or highly variable tasks), and 7) improving the pre-processing phase to include more robust feature extraction.

## 4. Conclusion & future directions

We conducted a feasibility study to demonstrate that combining ML-based image processing with B-mode ultrasound imaging can rapidly generate reliable fascicle length measurements for a large, superficial human calf muscle (*i.e.*, the soleus). Our results pave the way for applications that require non-invasive, real-time tracking of muscle state *in vivo*. Although marginally lower, correlation and RMSE values comparing ML-tracked to UltraTrack software derived ground truth, were comparable with those obtained through the most common post-processing approaches in the literature. There are a number of avenues to explore that could help refine the tracking performance of the current framework including optimizing ML algorithms to detect specific features in specific movement tasks or improving the ultrasound hardware itself. As ML-driven tracking of ultrasound images of human muscle continues to get faster and more accurate, post hoc tracking of muscle imaging data will be possible at high throughput and novel systems that employ muscle-in-the-loop feedback for biofeedback or wearable robotic control will enter the mainstream and form the basis for technologies that can continuously monitor injury risk, optimize muscle performance *in vivo*, and help improve mobility and quality of life.

## Supporting information

**S1 File. Supplementary material.**
(PDF)

**S1 Video. Hand-tracking demo.**
(MP4)

**S2 Video. UltraTrack demo.**
(MP4)

**S3 Video. Real-time demo.**
(MP4)

## Acknowledgments

The authors would like to acknowledge members of the Physiology of Wearable Robotics (PoWeR) Lab, Inan Research Lab (IRL), Dr. Owen Beck, and Nathan Glaser (All at Georgia Tech, US) for their help in polishing both concepts and code.

## Author Contributions

**Conceptualization:** Luis G. Rosa, Omer T. Inan, Gregory S. Sawicki.

**Data curation:** Luis G. Rosa.

**Formal analysis:** Luis G. Rosa, Gregory S. Sawicki.

**Funding acquisition:** Luis G. Rosa, Omer T. Inan, Gregory S. Sawicki.

**Investigation:** Luis G. Rosa, Gregory S. Sawicki.

**Methodology:** Luis G. Rosa, Jonathan S. Zia, Omer T. Inan, Gregory S. Sawicki.

**Project administration:** Luis G. Rosa, Gregory S. Sawicki.

**Resources:** Luis G. Rosa, Omer T. Inan, Gregory S. Sawicki.

**Software:** Luis G. Rosa.

**Supervision:** Omer T. Inan, Gregory S. Sawicki.

**Validation:** Luis G. Rosa, Gregory S. Sawicki.

**Visualization:** Luis G. Rosa.

**Writing – original draft:** Luis G. Rosa.

**Writing – review & editing:** Luis G. Rosa, Jonathan S. Zia, Omer T. Inan, Gregory S. Sawicki.

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
