## [Decision Letter · Decision Letter 0]

22 Feb 2021

PONE-D-21-01984

Machine learning to extract muscle fascicle length changes from dynamic ultrasound images in real-time

PLOS ONE

Dear Dr. Rosa,

Thank you for submitting your manuscript to PLOS ONE. After careful consideration, we feel that it has merit but does not fully meet PLOS ONE’s publication criteria as it currently stands. Therefore, we invite you to submit a revised version of the manuscript that addresses the points raised during the review process.

We look forward to receiving your revised manuscript.

Kind regards,

Yih-Kuen Jan, PhD, University of Illinois at Urbana-Champaign

Journal Requirements:

Reviewers' comments:

Reviewer's Responses to Questions

**Comments to the Author**

1. Is the manuscript technically sound, and do the data support the conclusions?

Reviewer #1: Yes

Reviewer #2: Yes

Reviewer #3: Yes

2. Has the statistical analysis been performed appropriately and rigorously? 

Reviewer #1: Yes

Reviewer #2: Yes

Reviewer #3: I Don't Know

3. Have the authors made all data underlying the findings in their manuscript fully available?

Reviewer #1: Yes

Reviewer #2: Yes

Reviewer #3: Yes

4. Is the manuscript presented in an intelligible fashion and written in standard English?

Reviewer #1: Yes

Reviewer #2: Yes

Reviewer #3: Yes

5. Review Comments to the Author

Reviewer #1: This paper aims to provide a benchmarking assessment of using several available machine learning programs to track soleus fascicle lengths during different functional movements, including calf raises and walking. The authors have made all data available and I believe this manuscript has broad impact in the fields of biomechanics and rehabilitation.

I do have a few comments for the authors to consider before final publication:

In lines 145-147 the authors mention that there was no movement of the heels or knees during constrained plantarflexion. However, it is extremely difficulty to provide a strong plantarflexor torque without movement in this position without the legs constrained (i.e. thighs fixed so knees cannot move). Was this the case? The images that show the different tasks don't look like this was the case. If so, please add.

In lines 197-199 - it is possible that I misinterpreted this analysis, but as lower RMSE would be better, I think there may be an error in your description on adding the inverse of RMSE (to make lower values better). Wouldn't inverting RMSE make higher numbers better since the usual RMSE would indicate lower is better? This is a bit confusing in this region so clarification would be appreciated.

Figure 4 - is the 7 seconds of data represented here standardized to the same portion of the gait cycle? Was RMSE/correlations different in various parts of the gait cycle? As indicated in the discussion, it would be expected if the correlations were higher during a portion of the gait cycle with less noise introduced by gait contact. Is this the case?

Reviewer #2: In this study, the authors have investigated the machine learning to extract muscle fascicle length changes from dynamic ultrasound images in real time. The topic of the manuscript is interesting. However, there are some minor concerns that should be addressed before the article can be accepted for publication.

1.Please describe the purpose of the study more specifically.

2.Please explain more about how the ultrasound images were obtained while the particitares rotate their ankle and walk task simultaneously?

3. In table 1 and 2, the parameter "CORR(r)" are calculated with other clinical data. Please describe how the values of "CORR(r)" are estimated with input data.

4.In table 2, what the possible reason for lower values of "CORR(r)" for cross-task training? And what's those values mean in the red color in table 1and 2?

5.It was mentioned six participants voluntarily participated in this experiment. Can the authors explain that the database is enough for training machine learning?

6.Which ML model is optimal by the results in this study?

Reviewer #3: The authors attempted to use machine learning (ML) models to record muscle fascicle lengths in real-time. The current result showed that the performance of support vector machine (SVM) model was the best with an acceptable r and RMSE, indicating that it is possible to track soleus muscle fascicle with ML models. However, there are still some issues that need to be clarified.

MAJOR COMMENT

Introduction

I was wondering why this study chose soleus as the target muscle. Could the authors explain why measuring soleus muscle fascicle length is important? For me, gastrocnemius may be more important than soleus in functional activities.

Methodology

First, how did you stabilize the ultrasound transducer on the muscle belly in performing dynamic tasks? There are various equipment or devices that stabilize the probe during dynamic movement in the past studies. Please elaborate this experimental setting in the method. Second, why did you ask participants to perform each movement in the same order? Did that affect the outcomes? Third, I did not understand the reason to select the last part of data as ground-truth rather than the random samples. Could you explain the explanation more? Why could this approach simulate the real-time measurement settings? Last, I did not understand how you calculated Pearson correlation coefficient. What 2 variables did you want to develop the correlation? How did you derive multiple r values to average them?

Results and Discussion

Could you separate the results from discussion? I still think it would be better to make 2 sections apart. In addition, I was curious why the SVM was the best model. Was there any possible reason?

MINOR COMMENT

Line 240: What does “moderately strong correlation” mean? I consider “moderate to strong” might be a better expression.

Line 383: “For example, it may be ‘that’ in…” What did “that” indicate in this sentence?

Line 404-406: “Furthermore, feeding ultrasound images…robustness and/or resolution.” Could you elaborate more why feeding data from other devices that assess muscle quality could improve the measurement robustness and resolution? In my opinion, these devices record different qualities of muscle, so they provide different information regarding muscle.

All tables: All tables need bottom border. And please put footnotes to explain CORR because r is not unit.

6. PLOS authors have the option to publish the peer review history of their article (what does this mean?). If published, this will include your full peer review and any attached files.

Reviewer #1: No

Reviewer #2: No

Reviewer #3: No

---

## [Author Response · Author response to Decision Letter 0]

12 Mar 2021

Thank you for your time and for considering our work. We have addressed all comments and look forward to working together towards a PLOS ONE submission. 

To see our detailed response to all comments, read our Response to Reviewers document attached in the submission website. *Might have to scroll down to the end of the document or end of the docs list to find it since we were unable to reorder documents. Thanks.

All Best.

---

## [Decision Letter · Decision Letter 1]

6 Apr 2021

PONE-D-21-01984R1

Machine learning to extract muscle fascicle length changes from dynamic ultrasound images in real-time

PLOS ONE

Dear Dr. Rosa,

Thank you for submitting your manuscript to PLOS ONE. After careful consideration, we feel that it has merit but does not fully meet PLOS ONE’s publication criteria as it currently stands. Therefore, we invite you to submit a revised version of the manuscript that addresses the points raised during the review process.

We look forward to receiving your revised manuscript.

Kind regards,

Yih-Kuen Jan, PhD

Academic Editor

PLOS ONE

Journal Requirements:

Reviewers' comments:

Reviewer's Responses to Questions

**Comments to the Author**

1. If the authors have adequately addressed your comments raised in a previous round of review and you feel that this manuscript is now acceptable for publication, you may indicate that here to bypass the “Comments to the Author” section, enter your conflict of interest statement in the “Confidential to Editor” section, and submit your "Accept" recommendation.

Reviewer #1: All comments have been addressed

Reviewer #2: All comments have been addressed

Reviewer #3: All comments have been addressed

2. Is the manuscript technically sound, and do the data support the conclusions?

Reviewer #1: (No Response)

Reviewer #2: Yes

Reviewer #3: Yes

3. Has the statistical analysis been performed appropriately and rigorously? 

Reviewer #1: (No Response)

Reviewer #2: Yes

Reviewer #3: Yes

4. Have the authors made all data underlying the findings in their manuscript fully available?

Reviewer #1: (No Response)

Reviewer #2: Yes

Reviewer #3: Yes

5. Is the manuscript presented in an intelligible fashion and written in standard English?

Reviewer #1: (No Response)

Reviewer #2: Yes

Reviewer #3: Yes

6. Review Comments to the Author

Reviewer #1: (No Response)

Reviewer #2: By the responses form the authors, it may be found that the comments have been addressed appropriately.

Reviewer #3: For the issue regarding movement task order, like you said, strenuous tasks might influence the muscle fascicle lengths. Therefore, it might be a limitation in this study.

7. PLOS authors have the option to publish the peer review history of their article (what does this mean?). If published, this will include your full peer review and any attached files.

Reviewer #1: No

Reviewer #2: No

Reviewer #3: No

---

## [Author Response · Author response to Decision Letter 1]

15 Apr 2021

Thank you all for your time and feedback. We have addressed the last comment on the Response to Reviewers Document and have updated the manuscript accordingly. Let us know if you still have any reservations and we will do our best to address them. Looking forward.

---

## [Decision Letter · Decision Letter 2]

21 Apr 2021

Machine learning to extract muscle fascicle length changes from dynamic ultrasound images in real-time

PONE-D-21-01984R2

Dear Dr. Rosa,

We’re pleased to inform you that your manuscript has been judged scientifically suitable for publication and will be formally accepted for publication once it meets all outstanding technical requirements.

Kind regards,

Yih-Kuen Jan, PhD, University of Illinois at Urbana-Champaign

Additional Editor Comments (optional):

Reviewers' comments:

Reviewer's Responses to Questions

**Comments to the Author**

1. If the authors have adequately addressed your comments raised in a previous round of review and you feel that this manuscript is now acceptable for publication, you may indicate that here to bypass the “Comments to the Author” section, enter your conflict of interest statement in the “Confidential to Editor” section, and submit your "Accept" recommendation.

Reviewer #3: All comments have been addressed

2. Is the manuscript technically sound, and do the data support the conclusions?

Reviewer #3: Yes

3. Has the statistical analysis been performed appropriately and rigorously? 

Reviewer #3: Yes

4. Have the authors made all data underlying the findings in their manuscript fully available?

Reviewer #3: Yes

5. Is the manuscript presented in an intelligible fashion and written in standard English?

Reviewer #3: Yes

6. Review Comments to the Author

Reviewer #3: (No Response)

7. PLOS authors have the option to publish the peer review history of their article (what does this mean?). If published, this will include your full peer review and any attached files.

Reviewer #3: No

---

## [Editor Report · Acceptance letter]

14 May 2021

PONE-D-21-01984R2 

Machine Learning to Extract Muscle Fascicle Length Changes from Dynamic Ultrasound Images in Real-Time 

Dear Dr. Rosa:

I'm pleased to inform you that your manuscript has been deemed suitable for publication in PLOS ONE. Congratulations! Your manuscript is now with our production department. 

Kind regards, 

on behalf of

Dr. Yih-Kuen Jan 

Academic Editor

PLOS ONE